# Novel Approaches to Target Mutant FLT3 Leukaemia

**DOI:** 10.3390/cancers12102806

**Published:** 2020-09-29

**Authors:** Jörg P. Müller, Dirk Schmidt-Arras

**Affiliations:** 1Institute of Molecular Cell Biology, Center for Molecular Biomedicine (CMB), Jena University Hospital, 07745 Jena, Germany; 2Institute of Biochemistry, Christian-Albrechts-University Kiel, 24118 Kiel, Germany; darras@biochem.uni-kiel.de

**Keywords:** acute myeloid leukaemia (AML), FMS-like tyrosine kinase 3 (FLT3), oncogenic signaling, re-sistance development, cancer cell vulnerability, haematopoietic niche

## Abstract

**Simple Summary:**

Acute myeloid leukemia (AML) is a haematologic disease in which oncogenic mutations in the receptor tyrosine kinase FLT3 frequently lead to leukaemic development. Potent treatment of AML patients is still hampered by inefficient targeting of leukemic stem cells expressing constitutive active FLT3 mutants. This review summarizes the current knowledge about the regulation of FLT3 activity at cellular level and discusses therapeutical options to affect the tumor cells and the microenvironment to impair the haematological aberrations.

**Abstract:**

Fms-like tyrosine kinase 3 (FLT3) is a member of the class III receptor tyrosine kinases (RTK) and is involved in cell survival, proliferation, and differentiation of haematopoietic progenitors of lymphoid and myeloid lineages. Oncogenic mutations in the FLT3 gene resulting in constitutively active FLT3 variants are frequently found in acute myeloid leukaemia (AML) patients and correlate with patient’s poor survival. Targeting FLT3 mutant leukaemic stem cells (LSC) is a key to efficient treatment of patients with relapsed/refractory AML. It is therefore essential to understand how LSC escape current therapies in order to develop novel therapeutic strategies. Here, we summarize the current knowledge on mechanisms of FLT3 activity regulation and its cellular consequences. Furthermore, we discuss how aberrant FLT3 signalling cooperates with other oncogenic lesions and the microenvironment to drive haematopoietic malignancies and how this can be harnessed for therapeutical purposes.

## 1. Introduction

Fms-like tyrosine kinase 3 (FLT3) is a member of the PDGFR (class III) receptor-tyrosine kinase (RTK) family and is expressed in human CD34+ haematopoietic stem cells (HSC), lymphoid progenitors, and progenitors cells of the granulocyte/macrophage lineage, including the common myeloid progenitor and the granulocyte/macrophage progenitors. As such, FLT3 participates in the maintenance of pluripotent HSC and contributes to proliferation and differentiation of B-cell progenitors, myelomonocytic and dendritic cells [1,2,3]. 

### 1.1. Classes of Activating FLT3 Mutations 

Overexpression of wild type (WT) or oncogenic forms of FLT3 have been implicated in several haematopoietic malignancies [4,5] and inflammatory disorders [6]. Mutations in the *FLT3* gene are present in approximately 30% of newly diagnosed AML [7]. Internal tandem duplication (ITD) mutations and tyrosine kinase domain (TKD) mutations are the most common classes of FLT3 abberations in AML. The overall incidence is approximately 23% for FLT3 ITD mutations and 7% for FLT3 TKD mutations [8]. The majority of ITD mutations occurs in the juxtamembrane region of FLT3, where it disrupts its autoinhibitory function (Figure 1). ITD mutations can also be located in the first kinase domain [8]. The ITD insertion site, which can vary in sequence and length is associated with resistance to chemotherapy and inferior outcome. Its integration site in the kinase domain was identified as an unfavorable prognostic factor for achievement of a complete remission, relapse-free survival, and overall survival of AML patients [9].

TKD mutations are predominantly found in the activation segment of the kinase domain and stabilise an active conformation of the activation segment (Figure 1). TKD mutations are most frequently observed at D835 and I836 with the substitution D835Y being the most frequently occurring TKD mutation. However, other substitutions within the activation segment were also reported. These substitutions stabilise an “open”, active conformation in which the activation segment is flipped out (Figure 1b), enabling access of ATP to the aspartate residue at position 829 that serves as a catalytic base [23]. Importantly, FLT3 TKD mutations display differential sensitivity towards TKI [24]. 

Upon TKI treatment, secondary mutations can occur, conferring TKI-resistance via alteration of the ATP/inhibitor-binding pocket. Additionally, these secondary mutations can, on their own, confer ligand-independent kinase activation. This has been in particular shown for the N676K substitution (Figure 1), which was first identified as TKI resistance-conferring mutation [12] and later established as an oncogenic driver mutation [25,26].

### 1.2. Biogenesis, Signalling and Regulation of the FLT3 Receptor Tyrosine Kinase

The FLT3 protein is co-translationally translocated into the endoplasmic reticulum (ER). Here, the luminal-faced N-terminus of the receptor undergoes multistep glycosylation and folding, as mediated by the ER luminal enzyme machinery [1]. The ER quality control system ensures that only properly folded FLT3 molecules egress the ER, get further glycosylated at the Golgi system to finally traffic to the plasma membrane (Figure 2a) [27,28]. 

There is an increasing body of evidence that activating mutations in receptor tyrosine kinases (RTK) cause aberrant intracellular localisation of these proteins. The reader is referred to a recent review [27]; which summarises molecular mechanisms and consequences of RTK mislocalization. Activating mutations in FLT3 were also shown to result in prolonged association of mutant FLT3 within the ER quality control and, therefore, resulting in a predominant ER localisation of FLT3 [28]. Aberrant activation of FLT3 at endomembranes results in altered downstream signalling quality [29,30]. In particular, the ER-resident pool of mutant FLT3 is responsible for phosphorylation and activation of STAT5 and was demonstrated to contribute to oncogenic transformation (Figure 2b) [29].

Constitutive phosphorylation of the receptor is a key determinant for the intracellular retention. An inactivating K644A point mutation of FLT3 ITD, treatment with FLT3 kinase inhibitors or overexpression of protein-tyrosine phosphatases promoted FLT3 surface localization [28,31]. Thus, in an “intracellular active kinase load” model Chan suggested that recruitment of phosphotyrosine-binding domain-containing proteins causes the retardation [32], but the molecular mechanism of FLT3 ITD retention in intracellular compartments is currently still not known. 

The dormant FLT3 resides in the cell membrane as an auto-inhibited monomer. Binding of its cognate ligand FL (FLT3 ligand) invokes conformational changes in the receptor ectodomain to establish a dimeric receptor assembly (Figure 2a). As a consequence of dimerization, the adjacent intracellular TKDs are trans-activated by auto-phosphorylation [33]. Stimulation of FLT3 mediates activation of several signal transduction pathways, including the mitogen-activated protein kinases ERK1/2 and the phosphoinositide-3-kinase/Akt signaling cascades (Figure 2a) [34,35]. 

Ligand-mediated phosphorylation of intracellular tyrosines induces FLT3 receptor endocytosis similar to that which has been observed for other RTKs such as the EGFR [36] or c-Kit [37]. FL-mediated internalization of both FLT3 WT and FLT3 ITD follows mainly a clathrin-dependent mode [38], and is associated with the induction of FLT3 degradation. Despite activity-independent internalization, ligand-induced receptor degradation depends on FLT3 kinase activity [38]. FLT3 WT and surface-bound FLT3 ITD exhibit similar ligand-induced internalization and degradation characteristics. 

Protein tyrosine phosphatases (PTP) were shown to predominantly antagonize FLT3 phosphorylation. Overexpression of ER-resident PTP1B [39] or perinuclear SHP-1 [40] resulted in dephoshorylation of oncogenic FLT3 ITD and stimulated its maturation [28]. Conversely, FLT3 interacting PTP SHP-2 was found to positively influence FLT3 signalling pathways [41,42]. Depletion or pharmacologic inactivation of SHP2 positively contributed to FLT3 ITD-induced haematopoietic progenitor hyper-proliferation and leukaemogenesis [43,44]. Expression of dual-specificity phosphatase (DUSP) 6 is up-regulated in FLT3 ITD positive cells and AML blasts, and was found to contribute to FLT3 ITD-mediated cell transformation [45]. Depletion of the membrane-localized receptor-type tyrosine-protein phosphatase eta (PTPRJ, Dep-1), and similarly receptor-type tyrosine-protein phosphatase C (PTPRC, CD45), resulted in enhanced phosphorylation and receptor-mediated downstream signalling activity of the FLT3 WT protein in myeloid cell lines. Direct interaction of PTPRJ and FLT3 was demonstrated by co-immunoprecipitation and in situ proximity ligation [46,47]. PTPRJ depletion stimulated colony formation of FLT3 ITD-expressing myeloid cells [48]. The absence of an antagonistic role of PTPRJ on FLT3 ITD was explained by oxidation of PTPRJ catalytic cysteines due to high FLT3 ITD-induced ROS levels [48]. Quenching of FLT3 ITD-mediated ROS-formation resulted in restoration of PTPRJ activity and subsequent inhibition of cell transformation [48]. Interference with NOX4-mediated overproduction of ROS resulted in PTPRJ re-activation in FLT3 ITD-expressing cells [37,38]. Breeding of FLT3 ITD knock-in mice [49] to *Ptprj*- or *Ptprc*-deficient mice, respectively, demonstrated that FLT3 ITD-signalling is controlled by PTPRJ and PTPRC in vivo (Figure 2b) [50,51]. Importantly, low level expression of both receptor PTPs correlated with a poor prognosis of FLT3 ITD positive AML patients [40,41]. 

Beside its effect on PTP activity, elevated ROS levels can also directly alter kinase function via cysteine oxidation as it was demonstrated for PDGFR [52], EGFR [53] and recently also for FLT3 [54]. Consistently, treatment of cells expressing FLT3 ITD with ROS-quenching agents attenuated signal transduction. Comprehensive analysis of cysteine-to-serine mutant FLT3 ITD proteins revealed critical roles of several cysteine residues for kinase activity and transforming signalling, further supporting cysteine modification as potential mechanism of activity regulation [54].

### 1.3. FLT3 Regulates HSC Self-Renewal and Aging 

All haematopoietic lineages propagate from a pluripotent HSC that exclusively possess the capacity for self-renewal to maintain life-long blood lineage replenishment. Ligand-mediated activation of FLT3 is one of the regulators of HSC self-renewal and differentiation [55,56]. In humans, FLT3 expression is detected on reconstituting short term (ST)-HSC, the Lin negative Sca-1^+^ c-Kit^+^ FLT3^+^ compartment [57], while there is no definitive evidence for its expression in human long-term (LT)-HSC [56]. In contrast in mice, *Flt3* transcripts and FLT3 protein were detected in the LT-HSC compartment suggesting a previously unrecognized role of FLT3 in LT-HSC homeostasis and establishes an intrinsic link between normal stem cell quiescence/homeostasis and development of myeloproliferative neoplasm [58].

While young LT-HSC have a quiescent cell cycle state and an unbiased differentiation capacity, their ageing results in enhanced proliferation and skewing of lineage commitment towards myeloid cells [59]. Loss of immune function and an increased incidence of myeloid leukaemia are two of the most clinically significant consequences of ageing of the haematopoietic system [60]. The NAD-dependent deacetylase sirtuin 7 (SIRT7) was shown to act as checkpoint for HSC maintenance. High SIRT7 level maintains HSC in a proper balance of quiescence, proliferation and lineage commitment. Aged HSC are specified by reduced SIRT7 [59,61]. Down regulation of SIRT7 in FLT3 ITD-expressing cells was recently established as relevant pathomechanism in AML [62]. Pharmacologic inhibition of FLT3 ITD or positive treatment response of patients restored *SIRT7* expression, suggesting that FLT3 ITD regulates HSC ageing and differentiation via SIRT7 [62]. The detailed molecular mechanism, how SIRT7 affects HSC differentiation and transformation in FLT3 mutant AML still remains elusive [63]. In contrast, overexpression of SIRT1 by a c-MYC-related network has been shown to contribute to the LSC maintenance in FLT3 ITD-positive AML [64].

## 2. FLT3 in Leukaemia

### 2.1. Initiating Events in Leukemogenesis

AML is characterized by clonal evolution and genetic heterogeneity of poorly differentiated cell clones derived from the haematopoietic system [65,66]. Several studies identified pre-cancerous signatures in exome data of healthy individuals [67,68,69]. More than 2% of healthy individuals (5–6% of people older than 70 years) contain mutations that may represent pre-malignant events linked to clonal haematopoietic expansion [69]. In particular, *DNMT3a*, *ASXL1* and *TET2* were significantly enriched for protein disruptive mutations. Strikingly, none of the samples analysed contained mutations in the proto-oncogene FLT3. This observation indicates that mutations in the *FLT3* gene are late events in the clonal expansion of haematopoietic stem and progenitor cells. 

### 2.2. Involvement of FLT3 in Other Haematological Diseases 

Mutations in the *FLT3* gene are not restricted to AML. Albeit uncommon in myelodysplastic syndromes (MDS), increased frequencies of *FLT3* mutations are associated with MDS progressing to secondary AML [70,71]. *FLT3* mutations are in general rare in acute lymphoblastic leukemia (ALL). Mutant *FLT3* and its overexpression were observed in Early T cell precursor T-lineage ALL [72] and Philadelphia chromosome-like ALL [73]. Adolescents and young adults with ALL also showed higher frequencies of *FLT3* mutations [74]. Zhang and co-worker even suggested the FLT3 pathway as potential therapeutic target for polycomb repressive complex 2 (PRC2)-mutated T-cell ALL [75]. To decipher the molecular mechanisms involved in the transition from the chronic phase to blast crisis in chronic myelogenous leukemia (CML), gene expression profiles at diagnosis in patients at the chronic phase and in blast crisis showed that increased abundance of *FLT3*-expressing cells attenuated imatinib-induced apoptosis [76]. 

### 2.3. Cell-Intrinsic Oncogenes Cooperating With Mutant FLT3 

*FLT3*, *DNMT3a* and *NPM1* are the most frequently mutated genes in cytogenetically normal AML [69,77]. Both, transgenic [49] and knock-in [78] mouse models expressing FLT3 ITD in the haematopoietic compartment revealed that FLT3 ITD mutations enhance survival and proliferation of lymphoid and myeloid progenitor cells and induced in particular a myeloproliferative syndrome [79], resembling chronic myelomonocytic leukaemia [78]. Thus, FLT3 ITD itself is not sufficient to induce AML in rodent and zebra fish models [80], but rather needs an initiating oncogenic event to fully drive leukemogenesis. Accordingly, in mouse models, FLT3 ITD was shown to cooperate with other oncogenic mutations found in patients to induce human AML-like syndromes. The earliest indication resulted from mice co-expressing FLT3 ITD with either partial tandem duplication of the gene encoding the histone methyltransferase mixed-lineage leukaemia (*MLL*) [81] or a MLL-AF9 fusion protein [82]. After a period of latency, these mice develop AML with short life span and extra-medullary involvement. Fusion of the nucleopore protein NUP98 to the homeobox protein HOXD13 occurs in human myelodysplastic syndrome [80]. Transgenic expression of NUP98-HOXD13 was shown in a mouse model to cooperate with FLT3 ITD to induce AML with short latency and 100% penetrance [83]. 

Mutations in the gene encoding the multifunctional nucleolar protein nucleophosmin (*NPM*) 1 occur frequently in myeloid neoplasia, disrupt its nuclear localisation and were shown in mouse models to develop late on-set leukaemia [80]. In mouse models, FLT3 ITD cooperates with *NPM1* mutations to result in rapid leukaemogenesis resembling human AML [84,85,86].

*DNMT3A* is a de novo DNA methyltransferase that is involved in CpG methylation and hence inactivation of genes, including maternal and paternal imprinting. Mutations in *DNMT3A* are preleukaemic lesions frequently mutated in apparently healthy persons. Thus, these mutations may represent characteristic early events in the development of haematologic cancers [67,68,87]. Genetic deletion of murine *Dnmt3A* along with subsequent expression of mutant FLT3 ITD in adult mice resulted in the development of acute myeloid and lymphocytic leukaemias [87]. Taken together, mutations of *FLT3* follow leukaemia-initiating events during the course of AML development.

### 2.4. Interaction of FLT3 Mutant Blasts With the Haematopoietic Niche

The physiologic HSC niche in the adult is located in the bone marrow and provides juxta- and paracrine signals to ensure maintenance and proliferation of HSC and to stimulate their differentiation into less potent, lineage-restricted progenitor cells. In particular, perivascular stromal cells play a crucial role for homing and residence of HSC by providing soluble and membrane-bound chemokine (C-X-C) ligand (CXCL) 12 (Figure 3a) and promote proliferation of HSC by the secretion of stem cell factor (SCF, encoded by *KITLG*). For further details on the cellular and molecular organisation of the haematopoietic niche, the reader is referred to recent excellent reviews [88,89].

In haematopoietic malignancies, the interaction of leukaemic cells with the haematopoietic niche is altered which contributes to pathology and therapy resistance. Patients suffering from haematopoietic malignancies including AML display loss of normal blood cell replenishment, which is associated with altered composition of the HSC compartment [90,91]. FLT3 ITD-induced myeloproliferation reduced the number of normal HSC in FLT3 ITD-transgenic mice and transplantation experiments. This effect was linked to enhanced expression of tumour necrosis factor (TNF) in endothelial cells and consequently in a reduction of mesenchymal stromal cells and endothelial cells (Figure 3b) [92]. Administration of etanercept, a recombinant dimerised version of TNFR2 ectodomain [93], which binds TNF and hence blocks TNF signalling, restored normal haematopoiesis in the mouse model [92]. Furthermore, there is strong evidence that FLT3 ITD-signalling enhances CXCR4 signalling in leukaemic cells [94,95]. FLT3 ITD-induced Pim1 kinase activity results in phosphorylation of CXCR4 intracellular domain and enhanced signalling of the chemokine receptor (Figure 3b) [96] and, therefore, enhances migration of FLT3 ITD^+^ leukaemic cells towards CXCL12 via enhanced Rho-associated kinase (ROCK) [97]. As a result, homing and residence of leukaemic cells to the HSC niche is enhanced.

The stromal cells of the haematopoietic stem cell niche were demonstrated to confer therapy resistance by different means. Recently, it was shown that bone marrow stromal cells express several cytochrom P450 isoforms, among them also CYP3A4 which was previously shown to be responsible for hepatic metabolism of TKI [98]. CYP3A4 in bone marrow stromal cells also contributed to resistance against three clinically used TKI (Figure 3c) [99]. Furthermore, stromal cells provide FL, which stimulates FLT3 expressed from the WT allele. Signalling of FLT3 WT on leukaemic blasts that also express a mutant FLT3 allele was shown to confer resistance to TKI [100].

In addition, bone marrow stromal cells were shown to confer TKI resistance [101] via provision of paracrine signals. IL-3 and GM-CSF were shown to rescue FLT3 ITD^+^ cells from tyrosine kinase inhibition (Figure 3c) [102] in vitro, which was at least in part via upregulation of the RTK Axl (Figure 3c and Figure 4d) [103]. However, FLT3 ligand (FL) and fibroblast growth factor (FGF) 2 were identified to be most potent in conferring TKI resistance and patients relapsing under TKI treatment demonstrated increased FGF2 expression in bone marrow stromal cells [104].

## 3. FLT3 as Therapeutic Target in Leukaemia

### 3.1. Small Molecule Inhibition of FLT3

Several small-molecule TKI targeting FLT3 have been developed and tested. (Recently reviewed in [105,106].) First-generation multi-kinase inhibitors (sorafenib, midostaurin, lestaurtinib) are characterized by a broad-spectrum of drug targets, whereas second generation inhibitors (quizartinib, crenolanib, gilteritinib) show more potent and specific FLT3 inhibition, and are thereby accompanied by less toxic effects. In 2017, midostaurin (PKC412) was approved by the FDA as first TKI to treat adult patients with newly diagnosed AML who are positive for oncogenic FLT3, in combination with chemotherapy [107]. The addition of midostaurin to standard chemotherapy significantly prolonged overall and event-free survival among patients with AML and a FLT3 mutation [108]. While Midostaurin acts rather unspecific on kinases in general, Quizartinib (AC220) inhibits selectively tyrosine kinases [109]. Phase I and II clinical trials have shown its survival benefit over conventional chemotherapy in patients with FLT3 ITD-positive relapsed/refractory AML [110]. Due to poly-pharmacology commonly observed for TKI, target space of approved or tested drugs were recently analysed for their target spectrum on human kinases. Here, cabozantinib, a TKI used to treat medullary thyroid cancer and renal cell carcinoma by targeting tyrosine kinases c-Met, VEGFR2, and also AXL and RET, was repurposed to abrogate oncogenic FLT3 ITD activity in AML patients [111].

### 3.2. Novel Alternative Approaches to Affect Oncogenic FLT3 Activity

Receptor dimerization is the prerequisite for FLT3 autophosphorylation. Thus, small-molecule inhibitiors of dimerization has become an attractive tool to interfere with protein–protein complex formation and subsequent down-stream signalling [112,113,114] that might open a new avenue the target FLT3. 

Enhancing enzymatic activity of counteracting PTPs is a potential rational approach to restrain oncogenic FLT3 kinase activity. As a proof-of-principle, PTPRJ activity was shown to be enhanced by Thrombospondin (TSP1) and to enhance dephosphorylation of its substrates [115,116]. Alternatively, interference with RPTP dimerization was identified to increase enzymatic phosphatase activity. Allosteric acting peptides preventing PTPRJ dimerization were recently demonstrated to reduce phosphorylation of EGFR and to antagonize EGFR-driven cell phenotypes [117] and might therefore represent a novel approach to regulate FLT3 activity (Figure 2b). Furthermore, chemical quenching of high ROS levels in FLT3 ITD positive cells were suggested to prevent reversible oxidation and consequently reactivate PTPRJ (Figure 2b) [118,119].

Statins were demonstrated to prevent complex FLT3 glycosylation resulting in loss of receptor surface localization and induction of cell death, as well as mitigation of TKI resistance [120]. Fluvastatin (Lescol), a generic drug approved in the USA in 2012, was shown to overcome resistance against the TKI sorafenib or the activation of the IL-3 compensatory pathway (Figure 4a). Important fluvastatin treatment in vivo reduced engraftment of BaF3 FLT3 ITD cells in Balb/c mice [120]. Combined inhibition of receptor glycosylation by fluvastatin or tunicamycin with TKI AC220 caused synergistic cell killing, which was highly selective for cell lines and primary AML cells expressing FLT3 ITD [121]. Synergistic effects observed in response to the combinatory application of pharmacologic inhibitors targeting FLT3 ITD downstream mediators are the basis for novel treatment strategies to deplete leukaemic cells. For instance, STAT5 inhibition synergistically increased cytotoxic effects of the JAK1/2 inhibitor Ruxolitinib or the p300/pCAF inhibitor Garcinol in leukaemic cells [122]. 

### 3.3. Exploiting Novel Vulnerabilities in FLT3 Mutant Blasts 

As indicated above, current therapies of FLT3 mutant haematopoietic malignancies involve the directed use of TKI. However, resistance development through either secondary FLT3 mutations or through the protection of FLT3 mutant cell clones by the bone marrow niche is a major obstacle for successful treatment. Therefore, novel therapeutic approaches are highly warranted. One attractive approach is to exploit novel vulnerabilities that are induced by mutant FLT3.

Activating mutations of FLT3 induce an unstable kinase conformation, resulting in the dependency on chaperones such as the heat shock protein (HSP) 90. Targeting HSP90 by geldanamycin and analogues has therefore been considered as TKI co-treatment that is also able to prevent TKI-resistance formation [123,124,125,126]. Novel therapeutic opportunities are also created through kinase-dependent impaired receptor maturation (see above), which has been recently reviewed elsewhere [27].

Expression of the protein arginine methyltransferase (PRMT) 1 is increased in AML blasts and it was demonstrated that in FLT3 ITD^+^ AML blasts, FLT3 is the major target of PRMT1. Genetic ablation of *PRMT1* increased apoptosis of FLT3 ITD^+^ blasts and prolonged survival of FLT3 ITD-expressing mice [127]. Consequently, pharmacological inhibition of PRMT1 decreased survival of FLT3 ITD^+^ blasts [127].

Recently, it was demonstrated that epidermal growth factor (EGF) receptor (EGFR) signalling enhances DNA damage repair (DDR) capacity in hepatocytes [128]. Apparently, RTK signalling is linked to DDR by a so far unknown mechanism. FLT3 ITD-signalling was also shown to be linked to DDR. In particular, expression of breast cancer-associated (BRCA) 1, which is part of a protein complex mediating homology-directed repair (HR) of DNA double strand breaks is enhanced in FLT3 ITD-expressing cells [129]. In contrast, efficiency of DNA double strand break repair by non-homologues end-joining (NHEJ) was decreased in cells expressing FLT3 ITD via down-regulation of Ku86 and up-regulation of the error-prone PARP1-dependent NHEJ pathway [130]. TKI-mediated FLT3 ITD inhibition results in a significant decrease of both, HR and NHEJ-mediated repair of DNA double strand breaks [129]. This seems to be conferred at least in part via Pim1 kinase activation [131]. Due to the resulting enhanced genotoxic stress, these cells were sensitized to PARP1 inhibition (Figure 4b) [129], similar as BRCA1-deficient mammary carcinoma cells [132]. 

Most likely due to its retention in intracellular compartments, FLT3 ITD was shown to be sensitive to the clinical approved proteasome inhibitor bortezomib [133], known to induce ER stress. The transcription factor ATF4 is present in ER stress and autophagy pathways (Figure 4b). It was demonstrated that FLT3 ITD induces autophagy and upon inhibition of autophagy, proliferation of leukaemic cells was impaired in vitro and in vivo [134]. 

A genome-wide CRISPR/Cas9 screen revealed that FLT3-ITD-expressing cells display alterations in their metabolic dependencies. Genetic deficiency in *GLS*, encoding for glutaminase, the first enzyme in glutamine catabolism, demonstrated to be synthetic lethal together with TKIs, indicating that FLT3-ITD-expressing cells maintain mitochondrial function and redox functions through glutamine catabolism (Figure 4c) [135]. Hence, the identification of novel synthetic lethalities, in particular, dependency on metabolic pathways and targeting key enzymes in these pathways represents a novel therapeutic approach that might synergize with tyrosine kinase inhibition.

Increased expression of the RTK Axl seems to be a current theme in therapy-resistant tumours [136,137,138,139], and was initially identified in EGFR inhibitor-resistant non-small cell lung carcinoma [139]. As already outlined above, FLT3-ITD-expressing leukaemic cells develop TKI-resistance via enhanced expression of Axl [140,141]. Up-regulation of Axl can also occur via reduced ectodomain shedding mediated by ADAM proteases [142]. Activation of Axl most likely occurs via autocrine secretion of its ligand Gas6 (Figure 4d) [140]. Resistance against FLT3 inhibitors was substantially reduced by co-inhibition of Axl [140,141]. Therefore, dual targeting of FLT3 and Axl seems to be a promising strategy to prevent TKI resistance and enhance therapeutic response. 

CAR-T cell technology has been developed to target malignant cell populations by direct killing of cells that expose a particular antigen. The prerequisite for successful targeted immunotherapies is the identification of suitable target antigens, which are restricted to malignant cells including malignant stem cells. For an excellent review on the state-of-the art of preclinical and clinical studies on suitable target antigens for CAR-T cell therapy in AML patients, see Hoffmann et al., [143]. Activated receptor tyrosine kinases are promising targets to defeat FLT3 ITD-positive AML blast cells. Since FLT3 ITD is retained in its biogenesis route (see above), TKI mediated receptor inactivation to promote FLT3 ITD surface localisation, is a prerequisite to target FLT3 ITD-positive cell clones efficiently. Combination of small molecule TKI crenolanib with CAR-T cells targeting FLT3 has been demonstrated as proof-of-concept that efficiency of CAR T-cell immunotherapy can be enhanced by the use of small molecule inhibitors. However, despite the curative potential of FLT3-CAR T-cell-based immunotherapies based on the graft-versus-leukaemia effect, severe side-effects have to be considered. Importantly, since FLT3-CAR T-cells recognize normal HSC, and consequently disrupt normal haematopoiesis, it has to be considered that adoptive therapy with FLT3-CAR T-cells will require subsequent CAR T-cell depletion and allogeneic HSC transplantation to reconstitute the haematopoietic system [144]. Furthermore, FLT3-CAR T-cell-based therapies have the danger of life-threatening graft-versus-host disease. Enhancing the graft-versus-leukaemia effect, but preventing graft-versus-host disease is an essential challenge for future therapy improvement [143]. 

### 3.4. Preventing Protective Signals from the Bone Marrow Niche 

Bone marrow stromal cells were shown to protect FLT3 ITD AML cells against TKI treatment by induction of enhanced STAT5 phosphorylation and enhanced AXL expression [103]. Thus, a further alternative therapeutic approach would be to target the protective bone marrow stromal niche and the interaction with the bone marrow niche. On one hand, inhibition of CYP3A4 has been shown to prevent metabolic TKI degradation. On the other hand, TKI resistance is also conferred by paracrine secretion of cytokines and growth factors such as IL-3, G-CSF or FGF-2, bypassing FLT3 inhibition by STAT5 and Ras/MAPK activation (Figure 4d). Some of these factors are synthesized as membrane-bound forms and released via limited proteolysis mediated by a disintegrin and metalloproteases (ADAM). Albeit, experimental evidence is still lacking, it is conceivable that inhibition of ADAM proteases could reduce TKI resistance formation. In contrast, membrane-bound proteins might also contribute to TKI resistance, as was previously suggested [145]. However, addition of the hypomethylating agent azacytidine was able to overcome TKI resistance in leukaemic blasts conferred by stromal cell contact [145].

## 4. Conclusions

Taken together, there are growing experimental data on how haematopoietic malignant cells exploit cellular components, signalling pathways and metabolic enzymes, resulting in relapse or refraction of targeted therapies. Combination of TKI-based therapy with targeting cooperating molecules or cells would result in synergistic effects to target resistance development and uncontrolled clonal development of leukaemic cells. Therefore, adopted clinical approaches synergistically attacking aberrant signaling events of leukaemic cells is the way to enhance therapeutic efficiency.

## Figures and Tables

**Figure 1 cancers-12-02806-f001:**
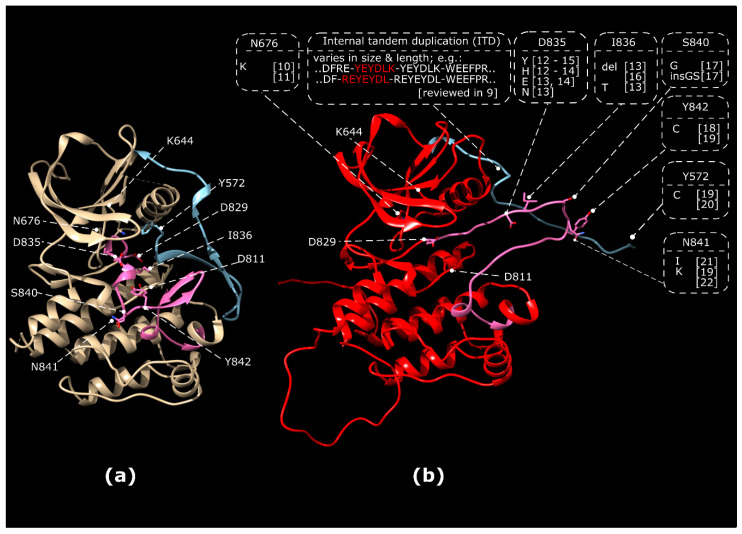
Activating ITD in the juxtamembrane domain and point mutations in the tyrosine kinase domain (TKD) of FLT3. (**a**) Inactive, autoinhibited conformation of FLT3 (1RJB.pdb). Residues involved in ATP-binding and catalysis (K644, D829) and residues subjected to mutation in leukemia are indicated. (**b**) Model of active FLT3 kinase domain demonstrating the region for ITD insertions and known frequent TKD point mutations resulting in activation of FLT3. Homology modelling of FLT3 kinase domain was based the active conformation of the CSF-1 receptor (3LCD.pdb) using MODELLER within the UCSF Chimera 1.14 software package. The activation segment of the kinase domain is marked in pink, the juxtamembrane domain is marked in blue. N676K [10,11], ITD reviewed in [9]. References: D835Y [12,13,14,15], D835H [12,13,14], D835E [13,14], D835N [13], D835V [13], I836 del. [13,16], I836T [13], S840G [17], S840insGS [17], Y842C [18,19], Y572C [19,20], N841I [21], N841K [19,22].

**Figure 2 cancers-12-02806-f002:**
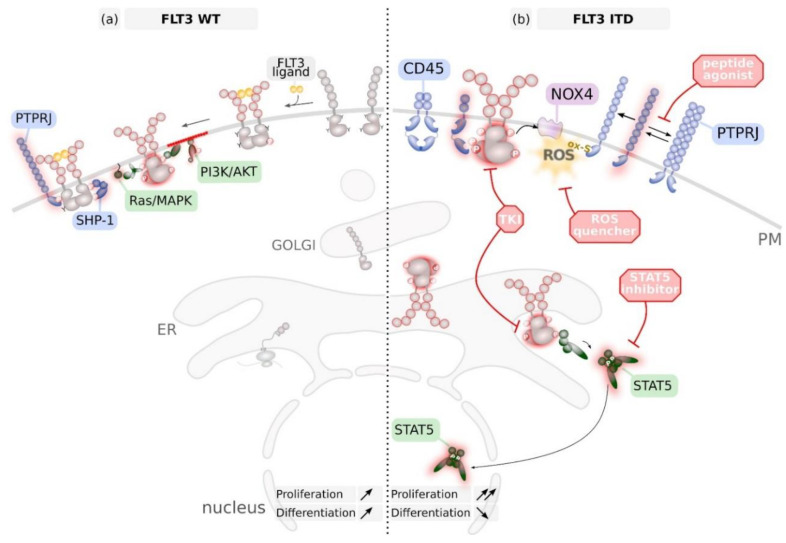
Biosynthesis and major signalling pathways activated by FLT3 wildtype (WT) (**a**) and FLT3 with internal tandem duplication (ITD) mutation (**b**). (**a**) Wildtype (WT) FLT3 is synthesized and processed in the endoplasmic reticulum (ER) and the GOLGI and reaches the plasma membrane (PM) as inactive monomer. Binding of the FLT3 ligand (FL) induces dimerization, autophosphorylation and induction of downstream signalling. This comprises activation of the Ras/mitogen activated protein kinase (MAPK) pathway and the phosphoinositol-3 kinase (PI3K)/AKT pathway from the PM and to some extent STAT5 phosphorylation from endosomes (not shown). Consequently, FLT3 WT signalling induces proliferation and differentiation of haematopoietic progenitor cells. FLT3 WT is inactivated via dephosphorylation by protein tyrosine phosphatases (PTPs) such as the transmembrane PTPRJ but also cytoplasmic PTPs such as SHP-1. (**b**) FLT3 with internal tandem duplication (ITD) mutations in the juxtamembrane region is constitutively and ligand-independently active at the PM and preferentially at endomembranes such as ER and endosomes (not shown). In particular, STAT5 is aberrantly activated at endomembranes. FLT3 ITD induces reactive oxygen species (ROS) production via activation of NADPH oxidase (NOX) 4. As a consequence, PTPRJ gets inactivated via oxidation of catalytic active cysteine residues. Dimerization of PTPRJ also reduces its catalytic activity. FLT3 ITD signalling can be blunted by the use of specific tyrosine kinase inhibitors (TKI) that are in clinical use. However, secondary mutations in FLT3 can cause inhibitor-resistance. Potential novel therapeutics comprise inhibitors of STAT5, ROS quencher and peptides that prevent dimerization of PTPRJ, thereby enhancing its activity.

**Figure 3 cancers-12-02806-f003:**
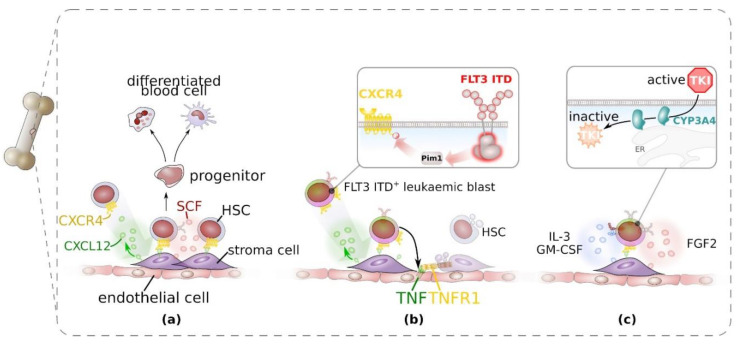
Interaction of haematopoietic stem cells (HSC) and FLT3 ITD-positive blast cells with the haematopoietic bone marrow niche. (**a**) Under physiological conditions, HSC are recruited and immobilised via the chemokine CXCL12, produced by bone marrow stromal cells in the perivascular niche. On HSC CXCL12 binds to and activates the G-protein coupled chemokine receptor CXCR4. HSC give rise to more restricted progenitor cells that fuel differentiated peripheral blood cells. (**b**) FLT3 ITD^+^ leukaemic blast cells shape the bone marrow haematopoietic niche. Through activation of the cytoplasmic serine/threonine kinase Pim1, FLT3 ITD enhances CXCR4 signalling and mediates leukaemic blast recruitment to the perivascular niche. Through the induction of tumour necrosis factor (TNF)-expression in endothelial cells, leukaemic blast cells drive TNF-mediated decay of stroma cells and in consequence, impaired normal haematopoiesis. (**c**) The perivascular niche, in particular stromal cells contribute to tyrosine kinase inhibitor (TKI)-resistance through (I) secretion of cytokines and growth factors that enable bypassing of inhibited FLT3 ITD and (II) CYP3A4-mediated catabolism of TKI in stromal cells.

**Figure 4 cancers-12-02806-f004:**
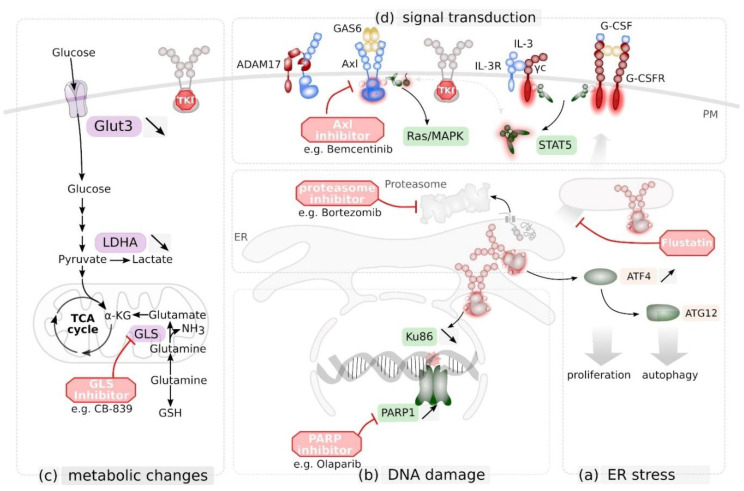
Targeting novel vulnerabilities in FLT3 mutant cells as novel therapeutic strategy. (**a**) FLT3 ITD induces expression of activating transcription factor (ATF) 4 that enhances the unfolded protein response (UPR) and triggers autophagy via autophagy related protein (ATG) 12. Export from the ER and degradation of misfolded proteins via the proteasome is part of the UPR. FLT3 ITD^+^ blasts are susceptible to proteasome inhibition via e.g. the clinically approved Bortezomib. Inhibtion of ER-to-Golgi trafficking via Flustatin further enhances UPR and UPR-mediated cell death in these cells. (**b**) FLT3 ITD signalling reduces Ku86 expression and consequently DNA damage repair by non-homologues end-joining. As a consequence, FLT3 ITD^+^ blasts rely on poly (ADP-ribose) polymerase (PARP) 1-mediated DNA damage response, making them vulnerable for PARP inhibitors such as Olaparib. (**c**) TKI-mediated inhibition of FLT3 ITD reduces expression of Glucose transporter Glut3 and lactate dehydrogenase (LDHA). As a consequence, ITD^+^ blasts rely on Glutamine as carbone source for the tricarboxylic acid cycle (TCA) cycle, making them vulnerable for glutaminase (GLS)-inhibitors such as the drug candidate CB-839 (**d**) Tyrosine kinase inhibitor (TKI)-mediated inhibition of FLT3 ITD signalling in leukaemic blast cells can be bypassed by expression and cytokine-mediated activation of the receptor tyrosine kinases Axl (by its ligand GAS6) or granulocyte colony stimulating factor receptor G-CSFR, or by activation of the interleukin (IL)-3 receptor complex. Signalling via Axl is regulated via ADAM17-mediated limited proteolysis. Simultaneous inhibition of FLT3 and Axl enhances therapeutic response. PM, plasma membrane; ER endoplasmatic reticulum.

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
