# Peer review of "Novel Approaches to Target Mutant FLT3 Leukaemia"

_cancers, 2020, doi:10.3390/cancers12102806_

Round 1

Reviewer 1 Report

The authors here give a well-structured and detailed summary of FLT3 function in HSC maturation and differentiation and deeply focused their attention in FLT3 physiopathological involvment in AML.

The review is well set out, with an “easy to read” structure and an appropriate bibliography.

I have only two suggestions:

- to deeply dissect the 3.3 chapter regarding the CAR-T cell technology: I would expect a more detailed pros and cons discussion on these point.

- to introduce a brief subchapter in section 2 (FLT3 in leukemia) describing its involvement also in other malignancies such as the blast crisis progression of CML.

Overall I recommend these review for publication after the implementation of the two proposed points.

Author Response

Dear Dr. Eneko Izaguirre,

Thank you very much for the revision of our manuscript. We highly appreciate the favourable comments from all three reviewers. We addressed all reviewers’ suggestions and comments. We hope that the manuscript in its current state is now suitable to be published in Cancers.

Please find below our detailed point-by-point responses to the issues raised by the reviewers.

Text modifications in the manuscript were highlighted in yellow.

Reviewer 1:

- to deeply dissect the 3.3 chapter regarding the CAR-T cell technology: I would expect a more detailed pros and cons discussion on these point.

In the original version of the manuscript, we only briefly discussed the possibilities to use CAR-T technology for the treatment of mutant FLT3 AML. As by the reviewer’s suggestion and due to the potency of the technique, we extended the text and discuss now in more detail the pros and cons:

CAR-T cell technology has been developed to target malignant cell populations by direct killing of cells that expose a particular antigen. The prerequisite for successful targeted immunotherapies is the identification of suitable target antigens, which are restricted to malignant cells including malignant stem cells. For an excellent review on the state-of-the art of preclinical and clinical studies on suitable target antigens for CAR-T cell therapy in AML patients, see Hoffmann et al., [131]. Activated receptor tyrosine kinases are promising targets to defeat FLT3 ITD-positive AML blast cells. Since FLT3 ITD is retained in its biogenesis route (see above), TKI mediated receptor inactivation to promote FLT3 ITD surface localisation, is a prerequisite to target FLT3 ITD-positive cell clones efficiently. Combination of small molecule TKI crenolanib with CAR-T cells targeting FLT3 has been demonstrated as proof-of-concept that efficiency of CAR T-cell immunotherapy can be enhanced by the use of small molecule inhibitors. However, despite the curative potential of FLT3-CAR T-cell-based immunotherapies based on the graft-versus-leukaemia effect, severe side-effects have to be considered. Importantly, since FLT3-CAR T-cells recognize normal HSC, and consequently disrupt normal haematopoiesis, it has to be considered that adoptive therapy with FLT3-CAR T-cells will require subsequent CAR T-cell depletion and allogeneic HSC transplantation to reconstitute the haematopoietic system [132]. Furthermore, FLT3-CAR T-cell-based therapies have the danger of life-threatening graft-versus-host disease. Enhancing the graft-versus-leukaemia effect, but preventing graft-versus-host disease is an essential challenge for future therapy improvement [131].

- to introduce a brief subchapter in section 2 (FLT3 in leukemia) describing its involvement also in other malignancies such as the blast crisis progression of CML.

We added a brief subchapter to the manuscript to give a short overview on how mutant and altered expression of FLT3 contribute to the development of other hematologic diseases:

2.2 Involvement of FLT3 in other haematological diseases

Mutations in the FLT3 gene are not restricted to AML. Albeit uncommon in myelodysplastic syndromes (MDS), increased frequencies of FLT3 mutations are associated with MDS progressing to secondary AML [60, 61]. FLT3 mutations are in general  rare in acute lymphoblastic leukemia (ALL). Mutant FLT3 and its overexpression were observed in Early T cell precursor T-lineage ALL [62] and Philadelphia chromosome-like ALL [63]. Adolescents and young adults with ALL also showed higher frequencies of FLT3 mutations [64]. Zhang and co-worker even suggested the FLT3 pathway as potential therapeutic target for polycomb repressive complex 2 (PRC2)-mutated T-cell ALL [65]. To decipher the molecular mechanisms involved in the transition from the chronic phase to blast crisis in chronic myelogenous leukemia (CML), gene expression profiles at diagnosis in patients at the chronic phase and in blast crisis showed that increased abundance of FLT3-expressing cells attenuated imatinib-induced apoptosis [66].

Reviewer 2:

Page 1, line 26 change expressed to expressed

The typo has bee corrected

Page 5, line 181-182, the interpretation of the paper referenced (59) is not fully correct. In the triple mutant, HLF is a crucial transcription factor, which modulates cell cycle dynamics and maintains the CD34+GPR56+ LSC-enriched compartment. The reference does not show that “co-occurrence of these mutations depends on the transcription factor hepatic leukaemia factor (HLF)” as stated in line 181-182, so please amend

We’re grateful to the reviewer to spot this incorrectness. The referenced paper does not show co-occurrence of these mutations. We corrected the phrase accordingly.

Page 5, line 191, MLL is a methyltransferase not an acetyltransferase, please amend

We apologize this incorrectness. MLL is encoding a histone methyltransferase. We corrected the sentence accordingly.

Page 6, line 202, I think that defining DNMT3A mutations as leukaemia initiating event is not correct as they are better defined as preleukaemic lesions, giving increased risk of developing leukaemia and which require cooperating mutation to fully develop leukaemia.

We agree with the reviewer that DNMT3A mutations are rather preleukaemic than a clear leukemia-initiating event as it has been stated by Genovese and Jaiswal. We changed the paragraph in the manuscript accordingly.

Mutations in DNMT3A are preleukemic lesions frequently mutated in apparently healthy persons. Thus, these mutations may represent characteristic early events in the development of hematologic cancers [57, 58][77].

Page 7 , pargraph 3.1 should mention other new FLT3 inhibitors above all the FDA approved and mostly used in clinic gilteritinib but also other in development such as crenolanib

According to the request of the referee, we extended the current TKI applications to target FLT3:

Several small-molecule TKI targeting FLT3 have been developed and tested. (recently reviewed in [95][96]. First-generation multi-kinase inhibitors (sorafenib, midostaurin, lestaurtinib) are characterized by a broad-spectrum of drug targets, whereas second generation inhibitors (quizartinib, crenolanib, gilteritinib) show more potent and specific FLT3 inhibition, and are thereby accompanied by less toxic effects. In 2017, midostaurin (PKC412) was approved by the FDA as first TKI to treat adult patients with newly diagnosed AML who are positive for oncogenic FLT3, in combination with chemotherapy [97]. The addition of midostaurin to standard chemotherapy significantly prolonged overall and event-free survival among patients with AML and a FLT3 mutation [98]. While Midostaurin acts rather unspecific on kinases in general, Quizartinib (AC220) inhibits selectively tyrosine kinases [99]. Phase I and II clinical trials have shown its survival benefit over conventional chemotherapy in patients with FLT3 ITD-positive relapsed/refractory AML [100]. Page 8, line 275 – correct awenue to avenue

The typo has been corrected

Page 9, line 325 – correct mamma to breast or mammary

The typo has been corrected

When discussing the role of sirtuins in FLT3 mutant AML a reference should be made to this paper too, Cell Stem Cell. 2014 Oct 2; 15(4): 431–446.

We added the 2014 identified role of SIRT1 on FLT3 to the manuscript:

Opposite, overexpression of SIRT1 by a c-MYC-related network has been shown to contribute to the LSC maintenance in FLT3 ITD-positive AML [54].

Reviewer III

This is an excellent review that briefly summarizes all aspects of aberrant FLT3 signaling. The authors provide basic information about activating FLT3 mutations and signaling events downstream FLT3. Part 2 discusses the role of FLT3 in AML development, whereas part 3 focuses on current and novel approaches to target oncogenic FLT3 activity. The manuscript is very informative for the broad audience, not just researchers with a primary focus on leukemia. A reference list covers the most relevant literature. In my opinion, the manuscript would be of high interest to the readership of Cancers. The only concern is the readability of the provided figures 2-4. Tiny fonts and color choices make the most of the text barely visible/hard to reads. My suggestion is to use more saturated colors to improve visibility.

The colors used for the figures have been intensified and text size of labels were increased to improve visibility of the Figures.

Reviewer 2 Report

This is a well written and overall comprehensive review of current knowledge of FLT3 biology in leukaemia and its targeting. The paragraphs cover all relevant points and figures are very explanatory. I have only few minor comments below:

Page 1, line 26 change expressed to expressed

Page 5, line 181-182, the interpretation of the paper referenced (59) is not fully correct. In the triple mutant, HLF is a crucial transcription factor, which modulates cell cycle dynamics and maintains the CD34+GPR56+ LSC-enriched compartment. The reference does not show that “co-occurrence of these mutations depends on the transcription factor hepatic leukaemia factor (HLF)” as stated in line 181-182, so please amend

Page 5, line 191, MLL is a methyltransferase not an acetyltransferase, please amend

Page 6, line 202, I think that defining DNMT3A mutations as leukaemia initiating event is not correct as they are better defined as preleukaemic lesions, giving increased risk of developing leukaemia and which require cooperating mutation to fully develop leukaemia.

Page 7 , pargraph 3.1 should mention other new FLT3 inhibitors above all the FDA approved and mostly used in clinic gilteritinib but also other in development such as crenolanib

Page 8, line 275 – correct awenue to avenue

Page 9, line 325 – correct mamma to breast or mammary

When discussing the role of sirtuins in FLT3 mutant AML a reference should be made to this paper too, Cell Stem Cell. 2014 Oct 2; 15(4): 431–446.

Author Response

(The authors gave the same response as above.)

Reviewer 3 Report

This is an excellent review that briefly summarizes all aspects of aberrant FLT3 signaling. The authors provide basic information about activating FLT3 mutations and signaling events downstream FLT3. Part 2 discusses the role of FLT3 in AML development, whereas part 3 focuses on current and novel approaches to target oncogenic FLT3 activity. The manuscript is very informative for the broad audience, not just researchers with a primary focus on leukemia. A reference list covers the most relevant literature. In my opinion, the manuscript would be of high interest to the readership of Cancers. The only concern is the readability of the provided figures 2-4. Tiny fonts and color choices make the most of the text barely visible/hard to reads. My suggestion is to use more saturated colors to improve visibility.

Author Response

(The authors gave the same response as above.)
